# AudioChat: Unified Audio Storytelling, Editing, and Understanding with Transfusion Forcing

**William Chen** [1 2]  **Prem Seetharaman** [1]  **Rithesh Kumar** [1 3]
**Oriol Nieto** [1]  **Shinji Watanabe** [2]  **Justin Salamon** [1]  **Zeyu Jin** [1]

## Abstract

Despite recent breakthroughs, audio foundation models struggle in processing complex multi-source acoustic scenes. We refer to this challenging domain as audio stories, which can have multiple speakers and background/foreground sound effects. Compared to traditional audio processing tasks, audio stories introduce new layers of semantic, temporal, and physical complexity. To address this challenge, we propose AudioChat, a framework for developing audio foundation models that can generate, edit, and understand audio stories. AudioChat introduces a new paradigm in which LLM-based toolcalling agents simulate interactions between users and the system, and these simulated dialogues are used as training data. We also introduce a novel Audio Transfusion Forcing objective to train the AudioChat model, allowing it to simultaneously decompose high-level instructions via structured chain-of-thought reasoning and perform interactive multi-turn audio understanding/generation. To evaluate generation and editing performance, we develop three new metrics that directly measure task performance instead of relying upon distribution-based scoring. We highly encourage readers to visit our demo to better understand the capabilities of AudioChat: https://wanchichen.github.io/audiochat/.

## 1. Introduction

Audio is a critical component in digital multi-media, acting as a foundational element for an immersive experience when consuming movies, podcasts, and audiobooks. Recent advances in foundation models (Valle et al., 2025; Evans et al., 2025; Défossez et al., 2024; Radford et al., 2023; Peng et al., 2023; Chen et al., 2025; Wang et al., 2023a) have demonstrated the remarkable ability of large-scale models in understanding and generating audio. However, these models focus on understanding and generating individual sounds, and are still limited in their ability to process complex acoustic scenes with multiple sound sources.

We refer to this frontier domain as audio story processing, which requires models to handle both non-speech and speech sounds with potentially multiple sources. It involves new challenges that are not found in typical text-to-speech (TTS), text-to-audio (T2A), and audio captioning (AC) tasks. Open-ended audio story editing represents the apex of such research, as it requires models to perform both audio story understanding and storytelling at a fine-grained level, such that only the desired elements to be edited are changed. Models must not only learn speech-specific details such as emotion and timbre, but also their relationship with non-speech sounds, such as ambience and pacing.

In this work, we present AudioChat, an audio foundation model that can perform open-ended text-guided audio understanding, storytelling, and editing on 48kHz polyphonic audio. AudioChat is capable of understanding, generating and editing complex acoustic scenes with ambient noise, both foreground and background sound effects, and multiple human speakers. Central to our approach is the concept of using *structured* chain-of-thought (CoT) reasoning (Wei et al., 2022; Nye et al., 2022; Jaech et al., 2024; Guo et al., 2025a) to break down abstract user inputs into individual sounds (Figure 1), allowing for interpretable processing.

We introduce several technical innovations to make the development of AudioChat feasible. We first create Audio-Copilot, a tool-calling Large Language Model (LLM) agent that simulates interactions between a user and a hypothetical AudioChat model, composing both the user instructions and desired output audio from scratch. We leverage AudioCopilot to simulate 6 million of these multi-turn conversations, which is then used to train the actual AudioChat model itself. Finally, AudioChat is trained with a novel Audio

[1]Adobe Research [2]Language Technologies Institute, Carnegie Mellon University [3]OpenAI. Work performed while at Adobe. Correspondence to: William Chen <williamchen@cmu.edu>, Prem Seetharaman <pseeth@adobe.com>.

*Proceedings of the 43$^{rd}$ International Conference on Machine Learning*, Seoul, South Korea. PMLR 306, 2026. Copyright 2026 by the author(s).

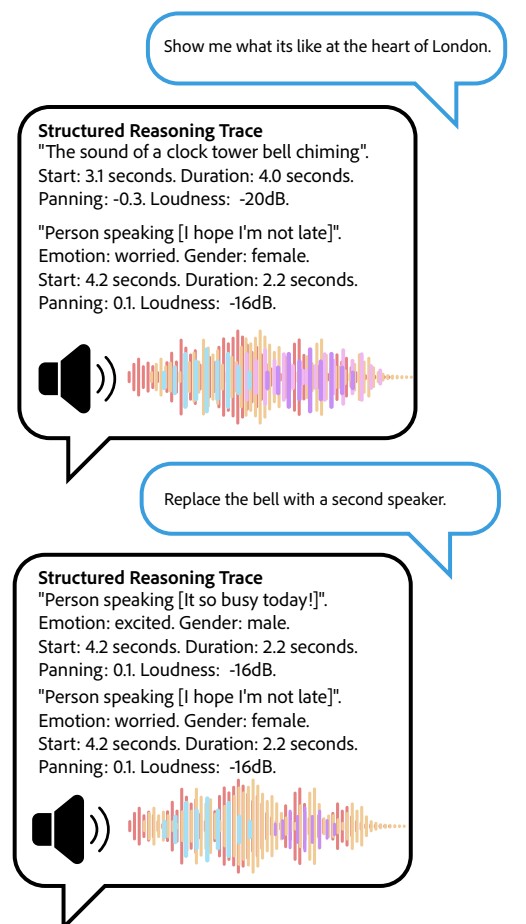

*Figure 1.* **Overview of AudioChat's capabilities in multi-source audio storytelling and editing.** AudioChat leverages structured CoT reasoning to break down the user prompt into individual sound effects, allowing for fine-grained control and interpretability.

Transfusion Forcing objective, allowing it to be optimized to directly perform structured reasoning, audio storytelling, and audio editing within a single end-to-end model. Our major contributions are as follows:

1. We develop AudioCopilot, a scalable pipeline for generating realistic audio scenes from scratch.

2. With the data generated from AudioCopilot, we develop AudioChat, the first unified model for generating, editing, and understanding audio stories with both speech and non-speech audio via structured reasoning.

3. We propose three novel evaluation metrics that overcome the weaknesses of standard audio generation metrics by directly measuring the task performance of audio generation and editing models.

## 2. Related Work

**Multi-Modal Language Models:** Multi-Modal LLMs (MLLMs) extend text-only LLMs to additional modalities,

such as images (Mo et al., 2025; Shi et al., 2024b; Deng et al., 2025) and audio (Chu et al., 2024; Xu et al., 2025a; Tian et al., 2025a). Architecturally, our work is most similar to that of Transfusion (Zhou et al., 2025) and BAGEL (Deng et al., 2025) from computer vision. We extend Transfusion to the multi-turn setting with our proposed Transfusion Forcing objective while maintaining a much slimmer design than BAGEL. In audio processing, TAC (Kumar et al., 2026) focuses on timestamped understanding of general audio events while AudioChat uses timestamp-level structured reasoning and understanding in the context of generation and editing. StreamRAG (Arora et al., 2025) takes the opposite approach to intelligent audio LLMs and instead optimizes for reduced latency in dialogue settings. Overall, Ming-UniAudio (Yan et al., 2025), UALM (Tian et al., 2025b) and Bagpiper (Tian et al., 2026) are the functionally closest to our work, developing unified MLLMs for both audio understanding and generation. The latter two require separate discrete audio tokenizers for understanding and generation (Chen et al., 2024b; Shi et al., 2024a), whereas AudioChat uses a simpler architecture with a single continuous audio tokenizer. MingAudio also uses continuous features for both understanding and generation, but only functions on speech. Similarly, UALM only supports non-speech audio and cannot generate intelligible speech. While Bagpiper and UALM use a similar reasoning-based method to ours, they require distillation from existing audio MLLMs. On the other hand, AudioChat represents a pipeline that is built independently from scratch. Furthermore, both UALM and Bagpiper can only perform high-level reasoning in natural language, while AudioChat's structured reasoning supports fine-grained controllability. Finally, none of these models consider spatial characteristics such as panning/loudness and are limited to 16kHz audio.

**Audio Storytelling:** Initial approaches for multi-source audio storytelling were primarily cascaded agent-based systems (Liu et al., 2025b; Xu et al., 2025b). These methods equip LLMs with single-task generative models (such as TTS or T2A) as tools and create the final audio scene by mixing the generated waveforms. However, the LLM does not have access to the generated audio, preventing any control and editing of the outputs. Agent-based systems also incur significant latency costs, since each sound is generated individually, which hurts their scalability to more complex scenes. Early end-to-end approaches like Make-An-Audio 2 (Huang et al., 2023) require fine-grained text inputs, limiting applicability. AudioStory (Guo et al., 2025b) builds upon this by using a reasoning LLM to generate sound timings. However, neither work can operate at the granularity of AudioChat - only supporting general directions (like "left" or "quiet") - nor generate stories with speech.

**Audio Editing:** Existing audio editing research has focused on models that only support a fixed set of operations, such

as insertion and deletion. This is accomplished by training models on random combinations of audio clips with known class labels (Wang et al., 2023b; Ellis et al., 2025). These works only use simple random mixtures of a few non-speech audio clips, limiting their effectiveness on more complex multi-source scenes. A recent work by Lan et al. (2025) proposes an approach similar to ours to support open-vocabulary editing. Given a user instruction, an audio-to-text model outputs a series of editing operations. For each instruction, a diffusion model outputs a new audio clip, which becomes the next audio clip to be edited. In comparison, AudioChat unifies the editing process into a single end-to-end approach within a simpler system design. This allows AudioChat to perform even more fine-grained editing, along with understanding and generation.

## 3. Audio Story Processing

### 3.1. Problem Formulation

Let $t$ be the number of model-user interactions. We formulate audio storytelling as modeling the joint probability of the structured reasoning chain $X_1^{\text{cot}}$ and the target audio $Y_1$ at $t = 1$, conditioned on the initial textual instruction $X_1^{\text{ins}}$. Using the product rule, we factorize this as:

$$P(X_1^{\text{cot}}, Y_1 | X_1^{\text{ins}}) = \underbrace{P(X_1^{\text{cot}} | X_1^{\text{ins}})}_{\text{Structured Reasoning}} \cdot \underbrace{P(Y_1 | X_1^{\text{cot}}, X_1^{\text{ins}})}_{\text{Audio Generation}}$$

(1)

Audio story understanding is formulated as the inverse problem. We assume the audio caption $X_t^{\text{cap}}$ depends primarily on the audio content $Y_t$ and is conditionally independent from $X_1^{\text{ins}}$. For an audio sequence $Y_t$, we model the probability of its corresponding caption $P(X_t^{\text{cap}} | Y_t)$.

Audio editing is the combination of both tasks. The model must modify a previous audio sequence $Y_{t-1}$ based on a new instruction $X_t^{\text{ins}}$. We formulate this as a recursive process involving three stages: understanding the input audio via captioning ($X_{t-1}^{\text{cap}}$), planning the modification via reasoning $X_t^{\text{cot}}$, and generating the edited audio ($Y_t$). The joint probability is factorized as:

$$P(Y_t, X_t^{\text{cot}}, X_{t-1}^{\text{cap}} | Y_{t-1}, X_t^{\text{ins}}) =$$
$$\underbrace{P(X_{t-1}^{\text{cap}} | Y_{t-1})}_{\text{Audio Understanding}} \cdot \underbrace{P(X_t^{\text{cot}} | X_{t-1}^{\text{cap}}, Y_{t-1}, X_t^{\text{ins}})}_{\text{Structured Reasoning}}$$
$$\cdot \underbrace{P(Y_t | X_t^{\text{cot}}, X_{t-1}^{\text{cap}}, Y_{t-1}, X_t^{\text{ins}})}_{\text{Audio Editing}}$$

(2)

For multi-turn editing (where $t > 1$), the caption of the previous audio $X_{t-1}^{\text{cap}}$ is equivalent to $X_{t-1}^{\text{cot}}$, the reasoning

chain used to generate it. This simplifies Equation 2 to:

$$P(Y_t, X_t^{\text{cot}} | Y_{t-1}, X_t^{\text{ins}}, X_{t-1}^{\text{cot}}) =$$
$$\underbrace{P(X_t^{\text{cot}} | X_{t-1}^{\text{cot}}, Y_{t-1}, X_t^{\text{ins}})}_{\text{Structured Reasoning}}$$
$$\cdot \underbrace{P(Y_t | X_t^{\text{cot}}, X_{t-1}^{\text{cot}}, Y_{t-1}, X_t^{\text{ins}})}_{\text{Audio Editing}}, \quad t > 1$$

(3)

The previous reasoning chain $X_{t-1}^{\text{cot}}$ effectively serves as the semantic state of the audio history. By carrying this context forward, the model maintains a continuous narrative, allowing it to generate the next reasoning step $X_t^{\text{cot}}$ without the computational redundancy of re-captioning the audio.

### 3.2. AudioCopilot

The primary challenge in developing models that can process complex audio scenes is the lack of training data, since it requires fine-grained annotations for each sound. Prior works rely on existing audio captioning models (Ghosh et al., 2025; Xu et al., 2025a), which cannot output at such granularity, or purely random mixing (Wang et al., 2023b; Huang et al., 2023), which has no semantic coherence.

To address this, we develop a synthetic data generation framework that exploits the synergy between audio storytelling, understanding, and editing. We use a novel tool-calling LLM agent, which we designate as AudioCopilot, to generate realistic story scenes from scratch as training data. We develop AudioCopilot by prompting a pre-trained LLM (Gemma 3 27B, (Team et al., 2025)) to simulate multi-turn interactions between a user and an AI sound designer, seeded by a random text string. The simulated user provides the text prompt for storytelling or editing, while the sound designer responds with the parameters needed to mix the scene together. These parameters are used to develop AudioChat's structured reasoning and audio story understanding capabilities. An overview is shown in Figure 2 and additional details can be found in Appendix A.

### 3.3. Training and Evaluation

**Training Data:** We simulate 6M conversations with Audio-Copilot as training data for AudioChat. Each conversation has 3 pairs of user text input/audio output. The average audio length is 8 seconds, leading to a total of 13.3K hours of data for storytelling and 26.6K hours for audio editing.

**Evaluation Data:** As there are no benchmarks for audio story processing, we develop StoryGen-Eval. StoryGen-Eval contains a curated set of 1200 simulated user-agent conversations generated with AudioCopilot by seeding LibriSpeech test-clean. Each conversation has 3 pairs of user-agent interactions, leading to a total of 1200 samples for storytelling/understanding and 2400 samples for audio edit-

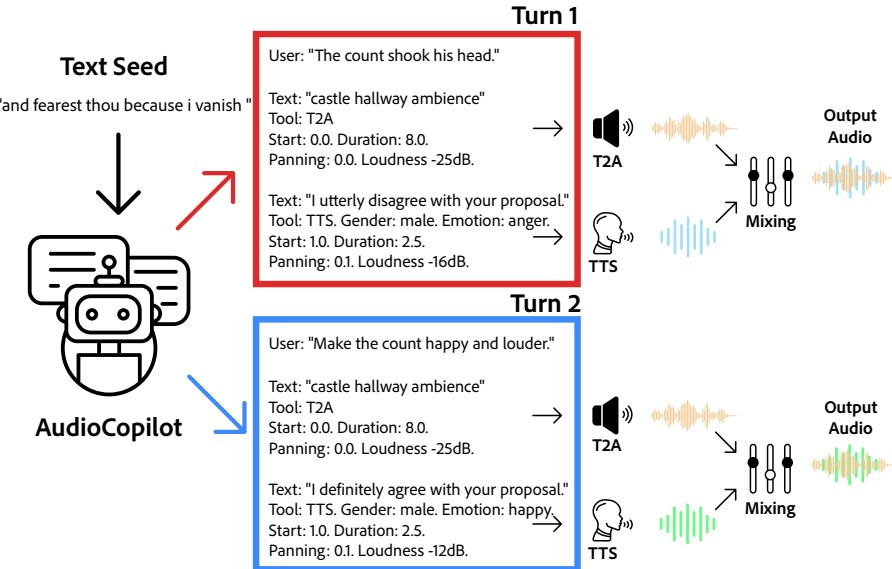

*Figure 2.* **Overview of data generation with AudioCopilot.** AudioCopilot seeds its dialogue simulation with a text string and performs tool-calling to render and mix each separate sound source.

ing. The editing samples are split among 6 tasks: open-ended editing, adding a sound, removing a sound, adjusting panning, adjusting volume, and changing the sound.

**Evaluation Metrics:** A major obstacle in developing audio story processing models is the lack of meaningful evaluation metrics. Commonly-used distribution-based approaches like Kernel Audio Distance (KAD) (Chung et al., 2025) and Fréchet Audio Distance (FAD) (Kilgour et al., 2018) cannot measure task-performance directly. Although CLAP-based metrics (Wu et al., 2023) are also often used for typical audio generation/understanding tasks, they cannot be directly applied to audio stories. Since CLAP is designed to match the entire audio clip to the entire text caption, the temporal resolution required to disentangle multiple sound sources is lost (Seki et al., 2025; Yuan et al., 2024).

One of our major contributions is the development of three new metrics that can be used to evaluate multi-source audio scenes: multi-caption FLAM score (multiFLAM), ΔmultiFLAM, and edit FLAM score (editFLAM). These metrics leverage OpenFLAM (Wu et al., 2025) a SOTA joint audio-text embedding model. Unlike typical CLAP-like models with pooled global representations, OpenFLAM outputs frame-level probabilities. This makes it significantly more robust in detecting if a sound occurred in an audio clip, which is important for fine-grained editing operations and better correlated with human judgment (Wu et al., 2025).

**multiFLAM** evaluates if a generated multi-source audio scene contains all necessary elements. Let $M$ captions be the set of captions of each sound source in $Y$. multiFLAM is calculated as the average probability of a sound correspond-

ing to caption $X_m$ occurring in scene $Y$ for all $m \in M$.

$$\text{multiFLAM}(X, Y) = \frac{\sum_{m=0}^{M} P_{\text{FLAM}}(X_m, Y)}{M} \quad (4)$$

We use multiFLAM to evaluate AudioChat's storytelling and understanding capabilities. For storytelling, it measures how likely the generated audio contains the sounds necessary to produce the story. For understanding, it measures the likelihood of each caption occurring in the input audio.

**ΔmultiFLAM**, or the change in multiFLAM scores across two audio inputs, can be used to evaluate audio editing. We use it to make sure all sounds meant to be unchanged in the input remain present in the edited output $\hat{Y}$. It is computed as the absolute value of the difference in multiFLAM scores across 2 input audio clips.

$$\Delta\text{multiFLAM}(X, \hat{Y}, Y) =$$
$$|\text{multiFLAM}(X, Y) - \text{multiFLAM}(X, \hat{Y})| \quad (5)$$

ΔmultiFLAM effectively measures the *consistency* of the edited audio with the original, given a set of audio captions.

**editFLAM** measures whether an audio editing operation itself was performed successfully. Generally, it is calculated as the difference in the FLAM probability between the input audio $Y$ and the predicted audio $\hat{Y}$. The text $e$ used to calculate the FLAM probability is the caption of the sound in $Y$ to be edited, $X_e$, where $e \in M$.

$$\text{EDITFLAM}(X, Y, \hat{Y}) = P_{\text{FLAM}}(X, \hat{Y}) - P_{\text{FLAM}}(X, Y) \quad (6)$$

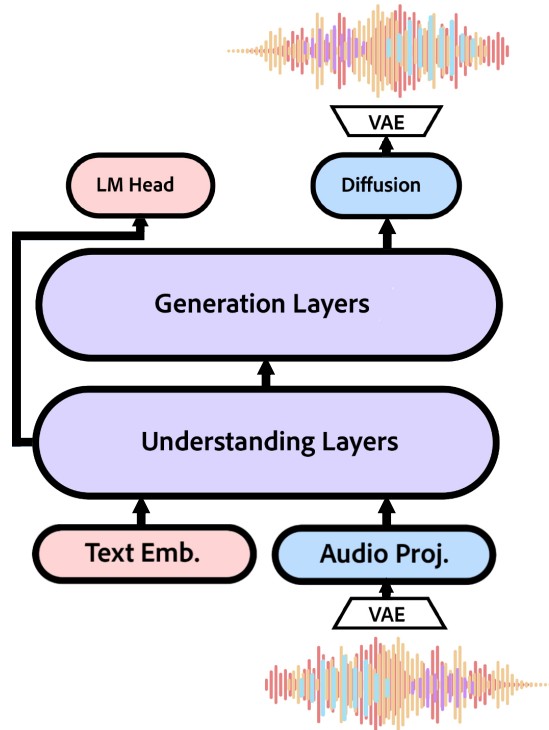

*Figure 3.* **Model architecture overview.** Tokens are input through a modality-specific tokenizer, the output of which is concatenated and input into the LM. The tokens are then routed through a modality-specific prediction head.

The exact implementation is dependent on the editing instruction. For example, when adding a sound, we instead compute $P_{\text{FLAM}}(X, Y) - P_{\text{FLAM}}(X, \mathbf{Y})$, so that higher is always better. This reference-free formulation allows editFLAM to be applied to a diverse number of open-ended editing tasks for real world audio. Further details for each metric are in Appendix E.

## 4. AudioChat

### 4.1. Architecture

AudioChat consists of 2 modules: a continuous audio tokenizer that converts the waveform into latent embeddings and a multi-modal LLM. An overview of this architecture is shown in Figure 3, whereas Section 4.2 further details the training procedure for the multi-modal LLM.

**Audio Tokenizer:** We use a continuous audio tokenizer, similar to DAC-VAE (Kingma & Welling, 2014; Kumar et al., 2023; Polyak et al., 2024), to convert an input waveform into continuous latent embeddings and vice versa. It receives stereo 48kHz audio as input and encodes it into a 40Hz latent embedding. The audio tokenizer is frozen during training. More details can be found in Appendix B.1.

**Multi-modal Language Model:** We introduce a novel architecture for injecting multi-modal information into text-

only LLMs, which we designate as the Self-Cascaded Transformer (SCT), inspired by recent advances in speech representation learning (Liu et al., 2025a; Chung et al., 2021). SCT sequentially splits the Transformer layers among two multi-modal tasks: understanding and generation. Given a $K$-layer Transformer, the first $U$ layers are trained only by a LM head (Section X) for understanding. The remaining $K - U$ layers are also optimized by the diffusion objective (Section Y) for generation. The text embeddings, understanding layers, and LM head directly correspond to and are initialized from a base text-only LLM. This design is significantly simpler and more flexible than previous proposed approaches like MoT (Liang et al., 2025), which require copying the entire base LLM's parameters and the use of multiple multi-modal representations (Deng et al., 2025). We empirically found that the SCT design significantly improved audio generation quality.

### 4.2. Audio Transfusion Forcing

We propose Audio Transfusion Forcing, a method to equip text-only LLMs with the ability to generate high-fidelity audio through continuous latent tokens within multi-turn interactions. Audio Transfusion Forcing consists of two key components: Transfusion and Diffusion Forcing.

**Transfusion:** Transfusion (Zhou et al., 2025) was originally proposed in computer vision to develop a model that can jointly perform generation and understanding tasks. It consists of two objectives: causal language modeling ($\mathcal{L}_{\text{LM}}$) and diffusion ($\mathcal{L}_{\text{DDPM}}$).

$$\mathcal{L}_{\text{TRANSFUSION}} = \mathcal{L}_{\text{LM}} + \lambda \mathcal{L}_{\text{DDPM}} \tag{7}$$

Where $\lambda$ is a weighting term to control for the variable dynamic range between the loss functions. Our use of Transfusion differs in two respects: 1.) we are the first to apply it to audio and 2.) in addition to joint generation/understanding, we use it to train a model that can decompose complex tasks via text-to-text reasoning.

**Language Modeling:** We adopt the framework of causal language modeling (Brown et al., 2020) to perform CoT reasoning conditioned on the user's input text. Given an $L$-length input sequence of discrete text tokens $X = (x_l | l = 1, ..., L)$, the LM is trained to model the product of conditional probabilities by optimizing the negative log-likelihood of $P(x_i | x_{1:i-1})$:

$$\mathcal{L}_{\text{LM}} = -\log \prod_{i=0}^{l} P(x_i | x_{1:i-1}) \tag{8}$$

Which yields the causal language modeling loss $\mathcal{L}_{\text{LM}}$ to be used in stochastic gradient descent.

**Diffusion:** Diffusion models (Ho et al., 2020; Song et al., 2021) are a class of generative models that learn to iteratively reverse a noise addition process. Briefly, the forward

process of diffusion is defined as a first-order Markov chain that gradually corrupts an input $\mathbf{y}$ with Gaussian noise over $T$ timesteps, yielding $\mathbf{y}_1, \mathbf{y}_2...\mathbf{y}_T$. The diffusion model thus learns the reverse process $P(\mathbf{y}_{t-1}|\mathbf{y}_t)$ by denoising the data at each timestep, where $t \in T$ is a randomly sampled timestep during training. In practice, this is accomplished by predicting the noise $\epsilon$ added to $y$ at timestep $t$, such that

$$\mathcal{L}_{\text{DDPM}} = ||\epsilon - \epsilon_\theta(\mathbf{y}_t, t, \mathbf{x})||^2 \qquad (9)$$

Where $\epsilon_\theta$ is the predicted noise as a function of $y_t$, $t$, and additional context $\mathbf{x}$, parameterized by the diffusion model. The overall optimization objective is therefore the mean-squared error between added noise $\epsilon$ and the predicted noise $\epsilon_\theta(\mathbf{y}_t, t, \mathbf{x})$. Equation 9 can then be generalized to multi-turn training for audio editing:

$$\mathcal{L}_{\text{DDPM\_MULTI}} = ||\epsilon - \epsilon_\theta(\mathbf{y}_t^s, t, \mathbf{x}, \mathbf{y}^0...\mathbf{y}^{s-1})||^2 \qquad (10)$$

Here, $S$ is the total number of conversation turns, $\mathbf{y}_t^s$ is the noised latent of the target audio at turn $s$ and $\mathbf{y}^0...\mathbf{y}^{s-1}$ are the clean audio latents from the previous turns.

**Diffusion Forcing:** One challenge of applying the above diffusion formulation (Equation 10) in the audio editing task is the extremely high correlation between the latent representations of the input audio context $\mathbf{y}^0...\mathbf{y}^{s-1}$ and the noisy target audio $\mathbf{y}_t^s$. This makes the training objective a near-trivial task, since the model can just simply copy the input latents, leading to poor generalization. While this training problem can be addressed by also noising the input audio (yielding $\mathbf{y}_t^0...\mathbf{y}_t^{s-1}$), this leads to a spurious correlation where the timestep $t$ is always the same for all $S$, breaking the inference process when $t_0...t_{s-1} = 0$ (no noise) and $t_s = T$ (full noise). To address this, we propose using Diffusion Forcing (Chen et al., 2024a), originally proposed for auto-regressive diffusion models, to independently noise $\mathbf{y}^0...\mathbf{y}^{s-1}$ and $\mathbf{y}^s$. Specifically, we independently sample a different timestep $t_s$ for each generation turn in $S$. Finally, this yields the overall audio generation objective by modifying Equation 10:

$$\mathcal{L}_{\text{DDPM\_FORCE}} = ||\epsilon - \epsilon_\theta(\mathbf{y}_t^s, t, \mathbf{x}, \mathbf{y}_{t_0}^0...\mathbf{y}_{t_{s-1}}^{s-1})||^2 \qquad (11)$$

Diffusion Forcing forces the model to perform audio generation with a variable amount of conditional audio information at each training step, mitigating both training collapse and train-test mismatches. This technique also allows us to simultaneously train the diffusion model with an arbitrary number of editing turns for each sample in parallel.

**Training Objective:** Combining Transfusion (Equation 7) and Diffusion Forcing (Equation 11) yields the overall Transfusion Forcing objective as a weighted sum of the text language modeling loss $\mathcal{L}_{\text{LM}}$ (Equation 8) and multi-turn diffusion forcing loss $\mathcal{L}_{\text{DDPM\_FORCE}}$ (Equation 11).

$$\mathcal{L} = \mathcal{L}_{\text{LM}} + \lambda \mathcal{L}_{\text{DDPM\_FORCE}} \qquad (12)$$

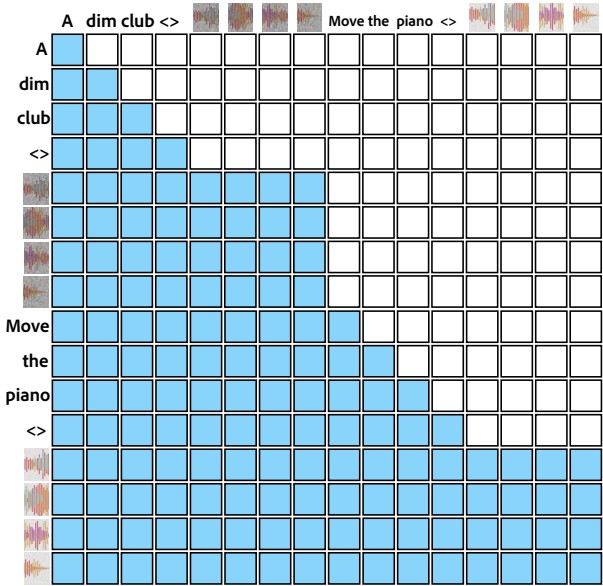

*Figure 4.* **Visualization of Transfusion Forcing:** a different diffusion timestep is sampled for each audio span. Audio tokens can attend to all other tokens within a span or in previous spans. Text tokens can only attend to previous text and audio tokens.

Which allows for the joint optimization of the model with a discrete distribution loss and continuous distribution loss in a multi-turn manner. We found $\lambda = 5.0$ to work well.

**Attention Masking:** We observed that naively maintaining the unidirectional attention of causal LMs leads to significantly degraded audio generation quality. While allowing for full bi-directional attention can ameliorate this issue, it prevents the model from performing multi-turn audio editing. We adopt a custom masking approach similar to (Zhou et al., 2025) and (Deng et al., 2025). We use causal attention for text, and bidirectional attention for audio within a single editing turn. Between editing turns, the audio attention is also causal: audio tokens in future editing turns can attend to audio tokens in previous turns, but not vice versa. This is visualized in Figure 4.

**Relationship to BAGEL:** Our architecture is inspired by that of BAGEL (Deng et al., 2025), which trains a vision LM via a similar transfusion forcing objective. One key difference lies in our simpler design - BAGEL shards the input tokens across modality-specific parameters. This introduces several constraints, such as the needing the same architecture across the text and diffusion tower. Our proposed SCT method removes this constraint, making the design more flexible. It also makes the implementation considerably simpler, since the LM's code does not need to be re-written to consider modality-specific packing, allowing for quick experimentation on different base LLMs. Another key change lies in the input representation format. AudioChat uses a unified audio representation while BAGEL uses multiple tokenizers and requires inputting both *clean and nosiy VAE*

*latents for the same image* during diffusion training. Further details are in Appendix B.2.

### 4.3. Multi-Stage Training

To amortize the cost of model training and development, we separate AudioChat's training into three main stages:

**Stage 1:** Starting from a text-only LLM (Gemma 2 2B (Gemma Team et al., 2024)), we initialize the additional DCT layers (yielding a total of 3.6B parameters) and jointly train on data for text language modeling, ASR (100K hours), audio captioning (10K hours), TTS (100K hours), and T2A (10K hours) for 100K steps.

**Stage 2:** We add the synthetic storytelling and editing data from AudioCopilot (Section 3.2). The model is trained for an additional 100K steps.

**Stage 3:** We retain most of the data used in Stage 2, but filter out lower quality synthetic AudioCopilot samples and train for a final 20K steps.

Additional details on the hyperparameters and data used in each stage are found in Appendix C.

## 5. Experiments

### 5.1. Audio Editing

**Setup:** As there are no publicly available audio editing models that can perform high-level audio editing, we establish controlled baselines to compare with AudioChat StoryGen-Eval. The baselines are used to evaluate each of the core methods we used to develop AudioChat: Transfusion Forcing, structured reasoning, and end-to-end training.

1. The DiT (Peebles & Xie, 2023) is used to evaluate the effectiveness of the Transfusion Forcing objective. It is conditioned solely on the edit instruction and original audio. Noise is not applied to the audio condition.

2. Diffusion LLM uses the same architecture as AudioChat, but is only trained with a diffusion loss and cannot perform chain-of-thought decomposition.

3. The cascade consists of an LLM and a storytelling model, which uses the same architecture as AudioChat. The LLM generates the CoT reasoning trace from the input edit instruction, which is synthesized into audio by the storytelling model. The storytelling model does not have direct access to the input audio.

All models are pre-trained and fine-tuned on the same data as AudioChat. Systems are evaluated along 3 axes: audio output quality (KAD and FAD), consistency with original audio (human evaluation and the change in multiFLAM ($\Delta$multiFLAM )), and instruction following (human evaluation and editFLAM). We ask 139 human annotators to rate

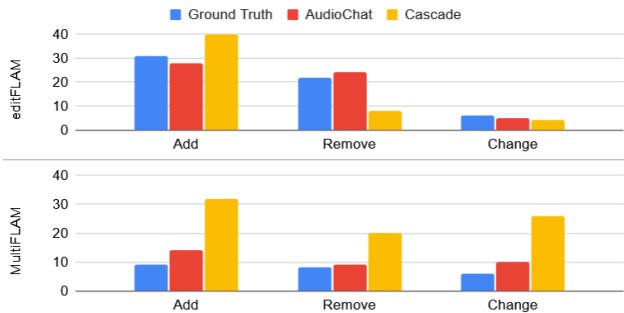

*Figure 5.* **editFLAM($\uparrow$) and multiFLAM($\downarrow$) results on the three semantic editing tasks in StoryGen-Eval.**

the systems on consistency and editing instruction following on a scale of 1-5 (higher is better), leading to 636 ratings per system. Further details are in Appendix E.

**Results:** Audio editing results are shown Table 1. AudioChat is the best performing system and obtains the best scores across all 6 metrics, showing the importance of each component in our model design. We also found that the DiT performs significantly worse than all other baselines. We attribute this to the stronger multi-modal representations via the MLLM architecture and our proposed Transfusion Forcing objective. Qualitatively, we observed that the DiT often copies the input audio without performing any editing, showing the importance of the diffusion forcing noise[1]. This is reflected in the evaluation: copying the input yields good consistency scores (26.7 $\Delta$multiFLAM) but poor instruction following scores (2.7 editFLAM). The cascade can achieve reasonable instruction following results (17.6 editFLAM) but struggles with not making extraneous changes (26.7 $\Delta$multiFLAM). This shows the importance of access to the audio context via end-to-end training, regardless of the details provided by the text CoT, and is best illustrated by Figure 5: the cascade performs well on adding sounds (which does not strictly require the input audio) but poorly on removal and change. Finally, our subjective results suggest that editFLAM correlates better with human judgment on the actual edit task ($\tau$=1.0) in comparison to FAD ($\tau$=0.3) at the system-level. $\Delta$multiFLAM also correlates with human judgment on adherence to the input ($\tau$=0.8), providing a more-interpretable alternative to FAD that works at the same-level.

### 5.2. Audio Storytelling

**Setup:** We compare AudioChat's audio storytelling capability with several open-source systems StoryGen-Eval, including Stable Audio Open (SAO) (Evans et al., 2025) and WavJourney (Liu et al., 2025b). Since SAO is not trained to

---

[1]We note that prior works have successfully leveraged DiTs for simpler audio editing tasks without diffusion forcing (Wang et al., 2023b). However, fine-grained audio story editing is a different challenge. Only a few sounds are different between input and output, leading to much higher correlation in the latent space.

*Table 1.* **Audio Editing Results:** Audio generation quality is measured in KAD/FAD, consistency with original audio via human ratings (1-5) and $\Delta$multiFLAM, and edit instruction following with human ratings (1-5) and editFLAM.

| | Difference | Quality | | Consistency | | Instruction | |
|---|---|---|---|---|---|---|---|
| | | KAD($\downarrow$) | FAD($\downarrow$) | Human($\uparrow$) | $\Delta$multiFLAM($\downarrow$) | Human($\uparrow$) | editFLAM($\uparrow$) |
| *Topline* | | | | | | | |
| Ground Truth | - | - | - | 4.12±0.05 | 8.2 | 3.68±0.06 | 19.4 |
| *Baselines* | | | | | | | |
| DiT | Architecture | 1.74 | 0.07 | 3.71±0.05 | 12.5 | 2.21±0.06 | 2.7 |
| Diffusion LLM | No CoT | 3.81 | 0.11 | 2.32±0.06 | 27.8 | 2.39±0.06 | 18.1 |
| Cascade | No audio context | 4.17 | 0.13 | 1.76±0.05 | 26.7 | 2.28±0.06 | 17.6 |
| *Proposed* | | | | | | | |
| AudioChat | - | **0.22** | **0.02** | **3.92** ± 0.05 | **11.7** | **3.12**±0.06 | **18.6** |

*Table 2.* **Audio Storytelling Results**

| | KAD($\downarrow$) | multiFLAM($\uparrow$) | Latency($\downarrow$) |
|---|---|---|---|
| SAO | 7.98 | 35.6 | **9.1** |
| LLM+SAO | 10.77 | 32.2 | 11.5 |
| WavJourney | 10.82 | **47.1** | 628 |
| AudioChat | **2.52** | 44.1 | 32 |

*Table 3.* **Fine-grained Audio Understanding results**

| | Diar. tcpWER | AC multiFLAM |
|---|---|---|
| Qwen2 Audio | 74.5 | 63,6 |
| AudioFlamingo 3 | 100+ | 76.6 |
| WhisperX | 55.9 | - |
| Whisper-Story | **5.5** | **88.1** |
| AudioChat | 9.7 | 86.3 |

process abstract user instructions, we create another baseline where an LLM (Gemma2 2B) first decomposes the instruction into individual sound effects, which are then generated (LLM + SAO). Systems are evaluated using KAD for audio output quality, multiFLAM for semantic consistency, and average latency in seconds on an A100 GPU.

**Results:** Table 2 shows the strong performance of AudioChat while incurring much lower latency costs compared to WavJourney, it has similar multiFLAM scores but 20 times faster. The poor results of both SAO-based systems show traditional T2A models cannot perform storytelling. For SAO, decomposing the abstract scene into individual sound effects with an LLM performed worse, likely since it was not trained on text prompts with many sound elements.

### 5.3. Audio Understanding

**Setup:** We evaluate AudioChat's fine-grained understanding capabilities on StoryGen-Eval. We use Time-Constrained Permutation Word Error Rate (tcpWER), the main evaluation metric for modern speaker diarization systems, with a collar of 1 second to evaluate fine-grained

speech understanding. For fine-grained non-speech audio understanding, we use our proposed multiFLAM metric. We use WhisperX (Bain et al., 2023) as the baseline system for fine-grained speech understanding. To evaluate AudioChat against dedicated audio story understanding models, we also create Whisper-Story by fine-tuning Whisper (Radford et al., 2023) on the synthetic story data from AudioCopilot, allowing it to also perform both tasks jointly. Finally, we also evaluate against two SOTA audio understanding models: Qwen2 Audio (Chu et al., 2024) and Audio Flamingo 3 (Ghosh et al., 2025).

**Results:** Audio story understanding evaluations are shown in Table 3. With a tcpWER of 9.7, AudioChat performs far better than the baseline WhisperX system (tcpWER of 55.9), despite the latter being a dedicated expert system for speech. This highlights the difficulty of the audio story understanding task and parallels findings on robust ASR (Cornell et al., 2023), further demonstrating the brittleness of standard speech understanding systems in complex acoustic environments. AudioChat only performs slightly worse than Whisper-Story overall on speech (5.5 vs 9.7 tcpWER) and comparably well on audio (0.88 vs 0.86 multiFLAM ), despite being pre-trained on less speech than Whisper (100K vs 5M hours) and being a joint generation/understanding model. A visualization is shown in Figure 7.

## 6. Analyses

### 6.1. Architecture

We perform a series of ablations on our design choices. We train different versions of AudioChat, differing only by the MLLM architecture: a vanilla dense model (no modification to LLM), MoT (Liang et al., 2025), and our proposed SCT method. For audio generation conditioned only on text, we test T2A performance using FAD and CLAPscore on AuditionSFX (Kumar et al., 2024). For text-audio conditioned audio generation, we use editing from our proposed StoryGen-Eval. We anchor our results with SOTA T2A

*Table 4.* **Architecture Ablations on Storytelling and Editing**

|  | T2A | | Editing | |
|  | FAD | CLAP | ΔmultiFLAM | editFLAM |
|---|---|---|---|---|
| SAO | 0.13 | 63.4 | - | - |
| TangoFlux | 0.23 | 56.1 | - | - |
| DiT | 0.14 | 61.6 | 12.5 | 2.7 |
| Dense | **0.12** | 65.3 | 15.5 | 17.2 |
| MoT | 0.13 | **65.8** | 16.4 | 9.6 |
| SCT | **0.12** | 65.5 | **11.7** | **18.6** |

*Table 5.* **Results on multi-turn editing:** There are only slight drops in performance when real generations used as input instead of the ground truth, indicating the robustness of our method.

|  | ΔmultiFLAM (↓) | editFLAM (↑) |
|---|---|---|
| Ground Truth (topline) | **10.5** | **11.1** |
| Previous Generation | 11.8 | 12.4 |

systems: SAO and TangoFlux (Hung et al., 2024). More information for each baseline is available in Appendix E.5.

The results in Table 4 verify the effectiveness of our methods. The SCT achieves strong performance compared on both tasks while being more lightweight than the MoT. While it achieves strong performance in T2A, we observed that the MoT struggles to "copy" the required sounds from the input for editing. We attribute this phenomenon to the weak cross-modal alignment of the MoT architecture - the model parameters are completely separate across modalities. The only cross-modal interaction in MoT is the parameter-free self-attention calculation (Appendix B.2).

### 6.2. Multi-turn Editing

In the previous section, we only evaluated a single editing turn. In this experiment, we examine the effectiveness of AudioChat after multiple rounds of editing. To do so, we evaluate the model's performance on two-turn editing (two input audios, one output audio) when given either the generated result from the previous turn or the ground truth audio on StoryGen-Eval. Table 5 shows that there is little degradation in performance when given the actual generation (about 1% absolute in both metrics), showing the robustness of our method to error cascades. We attribute this to AudioChat's long context capabilities - access to the original audio mitigates cascading errors from previous generations.

### 6.3. Real World Editing

Since AudioChat is only trained on synthetic editing data, we evaluate its robustness on real world audio from AudioCaps (Kim et al., 2019). To do so, we synthesize editing instructions using a pre-trained LLM (Yang et al., 2025) and evaluate AudioChat's outputs using editFLAM (ΔmultiFLAM is not possible since it requires fine-grained

*Table 6.* **Results when editing synthetic vs real world audio.**

| Test Set | Audio Type | editFLAM (↑) |
|---|---|---|
| StoryGen-Eval | Synthetic | 18.6 |
| AudioCaps | Real | 15.5 |

*Table 7.* **Comparison of different embedding models used for our proposed automatic evaluation metrics.** Rankings remain similar despite the change in the underlying model, showing the robustness of our metrics.

|  | FLAM | | PE-A | |
|  | Δmulti (↓) | edit (↑) | Δmulti (↓) | edit (↑) |
|---|---|---|---|---|
| Ground Truth | **8.2** | **19.4** | **2.7** | **23.6** |
| AudioChat | 11.7 | 18.6 | 3.0 | 23.3 |
| Diffusion LLM | 27.8 | 18.1 | 13.2 | 18.7 |
| Cascade | 26.7 | 17.6 | 19.1 | 23.1 |
| DiT | 12.5 | 2.7 | 4.1 | 4.0 |

captions). We observe that while there is indeed a decline in performance (18.6 → 15.5, Table 6), we believe drop is relatively small when considering that low-quality web audio is completely out of domain for AudioChat, which is only trained on high quality synthetic scenes. For a qualitative analysis, sample outputs can be found in our demo page.

### 6.4. Evaluation Metrics

Since our proposed automatic metrics rely on a pre-trained OpenFLAM embedding model (Wu et al., 2025), there is a risk of the results being skewed due to the underlying biases of OpenFLAM's training data. To control for this, we examine the difference in results when PerceptionEncoder-Audio (PE-A) (Vyas et al., 2025) is used as the embedding model. Our results in Table 7 shows that there is a difference in the dynamic range of the metrics, there is little change in the rankings of each method - the ground truth remains the topline, and thus still align well with the human evaluation scores in Table 1.

## 7. Conclusion and Future Work

We propose AudioChat, a multi-modal LLM that can generate, edit, and understand complex multi-source audio stories. To create the data necessary to develop AudioChat, we develop AudioCopilot, a tool-calling LLM agent that simulates interactions between users and an AI sound designer. AudioChat is trained with a novel Audio Transfusion Forcing loss, which efficiently combines structured CoT reasoning and multi-turn diffusion forcing. We evaluate AudioChat on a variety of and show that it can both generate and edit complex audio scenes, while being to perform fine-grained audio understanding. Finally, we propose 3 novel evaluation metrics that can better measure the performance of audio storytelling and editing models. In the future, we plan to inject AudioChat with visual capabilities for omni-modal understanding and generation.

## Impact Statement

This paper presents AudioChat, a foundation model that can understand, generate, and edit complex multi-source acoustic scenes. The goal of our research is to advance the field of multi-modal machine learning by developing audio processing models that can operate and be interpretable in a fine-grained manner. Since a portion of AudioChat's training data is proprietary, the model will not be publicly released in its current form. However, we believe that we have provided ample detail for reproducibility purposes, including prompts, dataset creation settings, training data statistics, hyperparameters, and ablation results. For example, the tools used by AudioCopilot can be easily replaced with similarly-performant open-source models (Casanova et al., 2024; Evans et al., 2025). The potential ethical considerations and social impact of AudioChat are similar to those of other audio generation models, which can be used for impersonation or fraud. We condemn such use and discourage any use of our findings for non-research purposes.

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

# A. AudioCopilot

AudioCopilot is our proposed LLM agent that we use for scalable data synthesis for audio stories. In contrast to prior work (Lan et al., 2025; Ellis et al., 2025; Huang et al., 2023; Wang et al., 2023b) that used mixes of real audio and text, we synthesize both the text and audio from scratch. This is done in order to generate realistic scenes with semantic coherence while not constraining the diversity of the generations. As shown in Figure 2, each generation of AudioCopilot is seeded from a text seed. For simplicity, we simply use the text transcripts from LibriSpeech (Panayotov et al., 2015) and the English portion Common Voice (Ardila et al., 2020).

## A.1. Architecture

AudioCopilot is built on top of a pre-trained Gemma 3 27B (Team et al., 2025). This model was selected due to its strong performance on text tasks while being small enough to perform inference on a single A100 GPU. For efficient generation, Gemma 3 is served using VLLM (Kwon et al., 2023). We experimented with other models, such as Llama 405B (Grattafiori et al., 2024) and Qwen3 235B (Yang et al., 2025), but did not observe substantial differences in output quality.

We equip AudioCopilot with two tools: a zero-shot TTS model and a T2A model. Both models operate at 48kHz, follow the latent diffusion formulation, and use the DiT architecture (Peebles & Xie, 2023). Data is generated by sampling from the TTS model for 64 steps, while the T2A model uses 24 steps.

## A.2. Data Generation Process

Given a text seed, we randomly choose one of two prompts for AudioCopilot (Figure 9). The first prompt gives it access to both the TTS and T2A tools, while the latter only provides access to T2A. We found this important for generating acoustically diverse samples. We qualitative improvements when the LLM generates an initial overview of the audio stories, before generating the description of each sound.

AudioChat will then generate a JSON corresponding to the audio story. Each story contains an array of sound elements. Each element contains parameters needed for tool usage (length and audio caption for T2A and length, transcript, emotion, and gender for TTS) and mixing (loudness, panning, start time). For TTS, we assign a random voice from a small internal dataset to each speaker in the JSON that matches the emotion and gender criteria. This assignment persists every time the speaker is present in the story, unless the user requests to change the gender. Finally, the sound descriptions are rendered into waveforms by the tools and then mixed together. To enforce editing consistency, AudioCopilot will only generate new sounds when the tool parameters are changed and not the mixing parameters. An example generation from AudioCopilot is shown in Figure 9.

## A.3. Data Quality

We experimented with different levels of data validation during the story generation process due to the inconsistency of LLM-based generation. We found that modern LLMs were sufficiently strong at generating realistic scenes in a one-shot manner without much prompt tuning. However, their performance was inconsistent at the O(10 million) scale. For example, we found that the LLM would change sounds that were not specified by the user or make changes to mixing parameters unprompted. As such, we implemented several prompts to ensure consistent generation quality.

1. Seeding from pre-existing text data (ASR transcripts, audio captions, etc.) was crucial for generating diverse scenes. Changing only the RNG seed with a fixed prompt led to small variations in the same few scenarios. This behavior has been studied in concurrent work on the lack of randomness in LLMs (Zhao et al., 2026).

2. Outlining exact boundaries for mixing parameter values, like -1.0 is hard left panning or -30dB is appropriate loudness for ambience. Otherwise, the LLM may generate objects that are too loud or too quiet.

3. Enforcing that ambience must be present throughout the entire audio. Otherwise, certain sounds may "hang" in isolation near the end of the audio.

Enforcing JSON-based validation helps address this, but it also causes a significant amount of generations to be filtered out. This validation was done by forcing the model to output which element IDs were added, removed, or changed across each editing turn (see Figures 9 and 10). We then compare those element IDs with the content of the generated JSON, filtering

out any mismatches. This process removed over 80% of the generated JSONs. Early models trained without the filtering could perform editing well (editFLAM of 17) but yielded poor consistency scores (20+ multiFLAM). As such, we adopted the pre-train + fine-tune approach discussed in 4.3 to leverage both large-scale weak data and the small high quality subset.

Finally, we note that we do not explicitly control for audio quality for cases where the T2A or TTS model may have failed (such as WER or CLAP filtering). These were very rare (upon a manual analysis of 100+ generated samples) and we believed could actually be helpful in certain editing cases (a sound could be changed even if the model mistakenly mis-captions one sound as another during inference). The samples used as editing input in the demo page are well-representative of the overall generated data.

Although we observed that training on all data without such validation worked well, it would cause models to make subtle texture and timing changes during editing. Only training on the small validated dataset was not sufficient for the model to learn the difficult tasks of storytelling and editing. We therefore used a multi-stage training setup to benefit from both settings, we first pre-train on all 6M samples and then fine-tune on the 100K validated samples.

## B. Architecture

### B.1. Tokenizer

We use a continuous audio tokenizer, similar to DAC-VAE (Kingma & Welling, 2014; Kumar et al., 2023; Polyak et al., 2024), to convert an input waveform into continuous latent embeddings and vice versa. The tokenizer receives stereo 48kHz audio as input and encodes it into a 40Hz latent embedding. Each audio channel is encoded independently into a 128-dimensional latent embedding. The channel-wise latents are then stacked along the hidden dimension, leading to a total embedding size of 256.

### B.2. Language Model

The LLM component of AudioChat totals to 3.6B parameters. It has 39 Transformer (Vaswani et al., 2017) decoder layers, each with a hidden size of 2304 and an MLP size of 9216. The model uses a vocabulary size of 256K with the weight tying between the LM head and input text embeddings. The first 26 decoder layers and the LM-head/text embedding weights are initialized from the 2.6B version of Gemma 2 (Gemma Team et al., 2024). We use the version that is only pre-trained on language modeling and is not instruction fine-tuned. A simple linear projection layer is used to map the 256-dim outputs of audio tokenizer to the Transformer's 2304 hidden size. Similarly, the diffusion head maps the 2304-dim Transformer output back to the 256-dim audio tokenizer space. We specifically use the $v$-pred formulation of diffusion training (Salimans & Ho, 2022).

As mentioned in Section 4.2, our design shares similarities to BAGEL (Deng et al., 2025). However, there are several key differences:

1. BAGEL uses three representations input representations per image: semantic tokens, clean VAE tokens, and noised VAE tokens. The latter is only used for diffusion training. These representations are concatenated along the time axis, leading to very long sequence lengths. AudioChat only uses noised VAE tokens for training input. During inference, the model uses clean VAE tokens for tasks requiring audio conditioning.

2. BAGEL uses the MoT architecture (Liang et al., 2025) by cloning the original LLM parameters. One set of parameters is used for understanding tasks, receiving semantic tokens as input. The other set of parameters is used for generation tasks, receiving VAE tokens as input. These two task-specific *towers* run in parallel, with the only cross-modal interaction occuring in the self-attention calculation. This is visualized in Figure 6 in our single tokenizer implementation of MoT. In comparison, the SCT uses a cascaded structure across modalities - understanding followed by generation. Not only is this more flexible, as the generation layers do not have to perfectly mirror the understanding layers, it is also more efficient. During text generation inference, only the understanding layers are activated.

While our initial experiments began with the MoT, we found that it performs poorly on interleaved generation tasks. As mentioned in Section 6.1, we attribute this to the weak cross-modal alignment mechanisms of MoT.

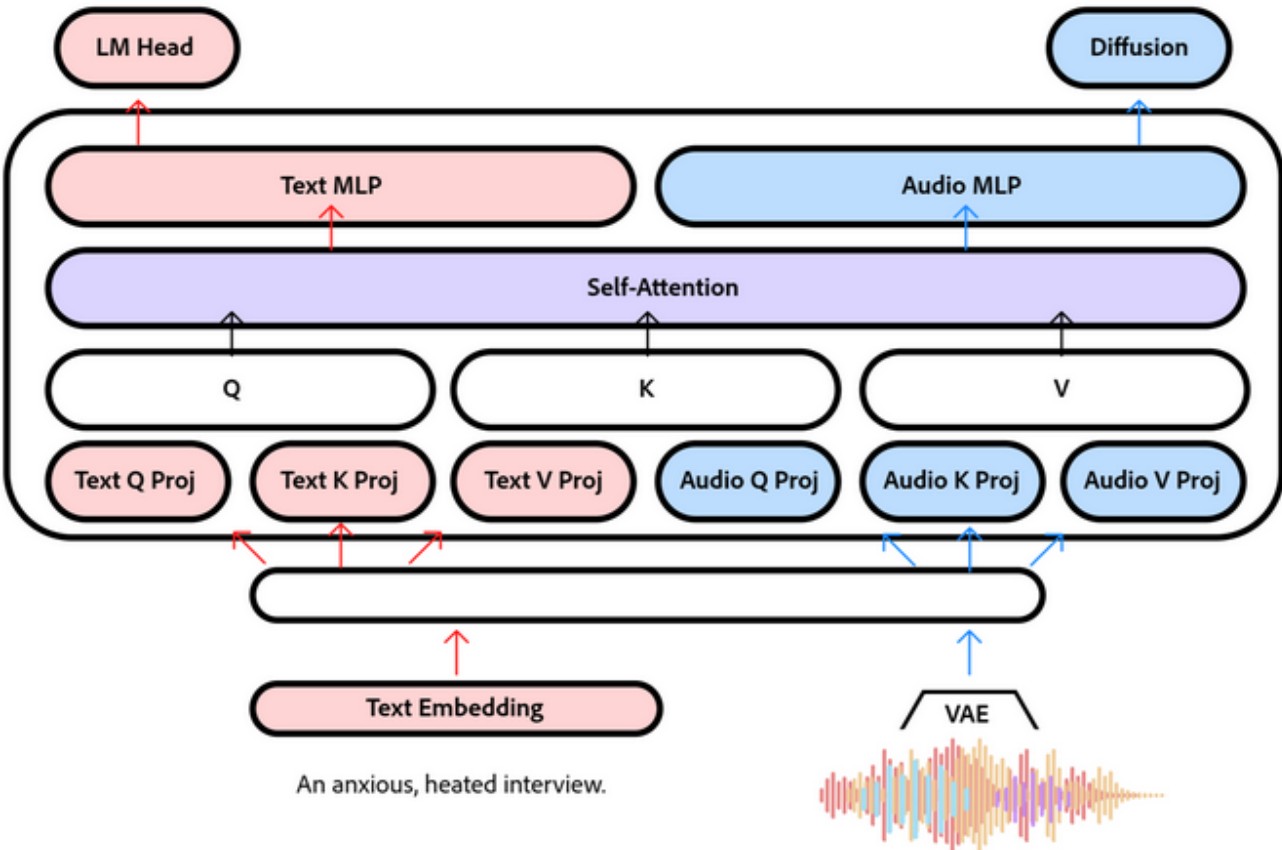

*Figure 6.* Overview of the MoT architecture.

## C. Training Settings

### C.1. Training Data

**ASR/TTS:** We use the same datasets for ASR and TTS, which total to 100K hours of English-only audio. Our data is mostly obtained from public sources, which are LibriLight (Kahn et al., 2020) and Common Voice (Ardila et al., 2020).

**AC/T2A:** We use the same datasets for audio captioning and text-to-audio, which total to 10K hours of non-speech data. 2K hours are obtained from Freesound, with music and non-commerical data filtered out. The remaining 8K hours are from internal and commercially-licensed datasets.

**TextLM**: To prevent the model from forgetting world knowledge learned from text, we perform continual text pre-training. We use the mid-training corpus from OLmO 2 (OLMo et al., 2024), which consists of high-quality documents for text language modeling. Following Tian et al. (2025a), we use the exact same sampling ratios of each split in the corpus as the original implementation (OLMo et al., 2024). While the original corpus contains over 300B tokens, in practice we only use 6B over the course of 200K model training.

**Audio Stories:** We use the 6M stories generated from AudioCopilot as training data for AudioChat, which corresponds to 13.3K hours for storytelling and 26.6K hours for editing.

**Audio Stories (High Quality):** We use the 100K validated AudioCopilt samples discussed in Section A.3, which corresponds to roughly 220 hours for storytelling and 440 hours for editing.

All data is resampled to 48kHz audio when training models. To train our models on fine-grained acoustic captioning, we also extract signal-level characteristics from the ASR/TTS and AC/T2A data, such as volume, panning, and duration.

*Table 8.* AudioChat Training Settings

|  | Stage 1 | Stage 2 | Stage 3 |
|---|---|---|---|
| Data Sources | ASR, TTS, AC, T2A, TextLM | Stage 1 + Stories | Stage 1 + Stories HQ |
| *Optimization* |  |  |  |
| Training Steps | 100K | 100K | 20K |
| Optimizer | AdamW | AdamW | AdamW |
| Learning Rate | 2.5e-5 | 2.5e-5 | 1.0e-4 |
| Scheduler | Cosine Decay | Cosine Decay | Cosine Decay |
| Warmup Steps | 500 | 0.0 | 2500 |

### C.2. Hyper-parameters

Training hyper-parameters are shown in Table 8. Unlike previous works on training MLLMs (Deng et al., 2025; Ghosh et al., 2025), we found that an alignment pre-training stage with the audio tokenizer and LLM frozen was not necessary if the audio embedding norms were scaled to the same magnitude as the text embedding norms. We found this important for learning on audio-to-text tasks. Otherwise, the model would learn to ignore the audio. Interestingly, the scaling was not necessary for text-to-audio tasks. We leave further exploration of this to future work.

## D. Inference Settings

This section details the inference parameters we used to obtain our results. For audio editing and storytelling (Sections 5.1 and 5.2), we sample with 150 diffusion steps. For our T2A ablation results, we use 100 diffusion steps for all models. For all audio generation tasks, we use Classifier-Free Guidance (Ho & Salimans, 2021) with a weight of 3.0, the Karras noise schedule (Karras et al., 2022) and DPM-Solver++ (Lu et al., 2025). We use greedy search for all text generation results.

## E. Evaluation Settings

### E.1. multiFLAM

multiFLAM evaluates if a generated multi-source audio scene contains all necessary elements. Let there be $M$ captions, one for each sound source in $Y$. multiFLAM is calculated as the average probability of a sound corresponding to caption $X_m$ occurring in scene $Y$ for all $m \in M$. First, we use FLAM to output the probabilities of $X_m$ for each timestep $y_t \in Y$, which are then aggregated by taking the maximum value over $T$.

$$P_{\text{FLAM}}(X_m, Y) = \text{MAX}(P_{\text{FLAM}}(X_m, y_t)|t = 1, ...T) \tag{13}$$

We use the maximum instead of mean since the latter unfairly punishes detecting sounds that only occur in small portions of the audio. The average probability across sounds are then calculated following Equation 4, yielding the final result.

### E.2. ∆multiFLAM

∆multiFLAM evaluates the consistency an editing operation to its original audio. Given the sound to be edited $e \in X$, we calculate the multiFLAM of $Y$ and $\hat{Y}$ using $X' = X - \{e\}$.

$$\Delta\text{multiFLAM}(X', \hat{Y}, Y) =$$
$$|\text{multiFLAM}(X', Y) - \text{multiFLAM}(X', \hat{Y})| \tag{14}$$

In other words, ∆multiFLAM is the change in multiFLAM score for all sounds that should **not** be edited. Scores closer to 0 indicate better consistency with the original, whereas scores closer to 1.0 indicate more unnecessary changes.

### E.3. editFLAM

editFLAM measures whether an audio editing operation itself was performed successfully. Generally, it is calculated as the difference in the FLAM probability between the input audio $Y$ and the predicted audio $\hat{Y}$ (Equation 18). As mentioned in Section 3.3, the exact implementation depends on the task. In this section, we provide the implementation of each task that we evaluate using the metric.

**Adding a sound**: When adding a sound, the probability of caption $X$ occurring in $\hat{Y}$ should be higher than the probability of it occurring in $Y$:

$$\text{EDITFLAM-ADD}(X, Y, \hat{Y}) = P_{\text{FLAM}}(X, \hat{Y}) - P_{\text{FLAM}}(X, Y) \tag{15}$$

A positive score indicates an absolute increase in the probability that the sound was successfully added. A score around 0 indicates that the sound was likely not added. A negative score indicates unintended distortions in the edited audio.

**Removing a sound**: When removing a sound, the probability of caption $X$ occurring in $\hat{Y}$ should be lower than the probability of it occurring in $Y$:

$$\text{EDITFLAM-REMOVE}(X, Y, \hat{Y}) = P_{\text{FLAM}}(X, Y) - P_{\text{FLAM}}(X, \hat{Y}) \tag{16}$$

To normalize each task score so that higher is better, we invert the subtraction. A positive score indicates an absolute increase in the probability that the sound was successfully removed. A score of around 0 indicates that the sound was likely not removed. A negative score indicates unintended distortions in the edited audio, which may include making the target sound more prominent.

**Changing a sound**: One of the most complex operations to evaluate in editing is changing a sound $X_1$ to $X_2$. We formulate this as a combination of two sub-tasks: removing $X_1$ and adding $X_2$.

$$\text{EDITFLAM-CHANGE}(X_1, X_2, Y, \hat{Y}) = \frac{\text{EDITFLAM-ADD}(X_2, Y, \hat{Y}) + \text{EDITFLAM-REMOVE}(X_1, Y, \hat{Y})}{2} \tag{17}$$

The final score is the average of the two sub-task scores. A positive scores indicates that both changes were likely to have occurred successfully. Scores closer to 0 indicate that there was likely no change in the sound. Negative scores indicate unintended distortions in the edited audio.

**Changing acoustic level of a sound**: When changing the acoustic level of a sound (such as volume or panning) and not the semantics, the probability of caption $X$ occurring in $\hat{Y}$ should be no different than the probability of it occurring in $Y$:

$$\text{EDITFLAM-ACOUSTIC}(X, Y, \hat{Y}) = 1.0 - |P_{\text{FLAM}}(X, \hat{Y}) - P_{\text{FLAM}}(X, Y)| \tag{18}$$

To normalize the task score so that higher is better, we compute the 1.0 minus the absolute value in the change in probability. A higher score indicates that it is likely that no semantic changes were made to the sound, while scores closer to 0 indicate significant changes. We note that this formulation will also punish models that lower the volume of a sound to 0, effectively removing the sound, which we observed to be a common shortcut taken by models.

### E.4. Human Evaluation

We performed subjective human evaluation on the edited audio story outputs. Annotators were given the input audio, the edit instruction, and audio output from one of the systems (Ground truth audio, DiT, Diffusion LLM, Cascade, and AudioChat). The annotators were asked to rate the audio edit along two axes:

**Consistency with input audio (1-5)**: Aside from the specified changes in the instruction, is the output audio consistent with the original audio mix? Please only consider the consistency with the original audio and do not consider the instruction following criteria when rating for consistency.

- 1: the output audio does not resemble the original audio mix at all.

- 2-4: the output audio resembles the original audio mix, but there are some differences.

- 5: Outside of the specified changes, the output audio sounds exactly like the original audio mix.

**Instruction following (1-5)**: Is the output audio correctly changed following the text instruction? Please use the text instruction to guide your judgment.

- 1: The output audio does not follow the text instruction at all.

- 2-4: The audio is changed somewhat according to the text instruction.

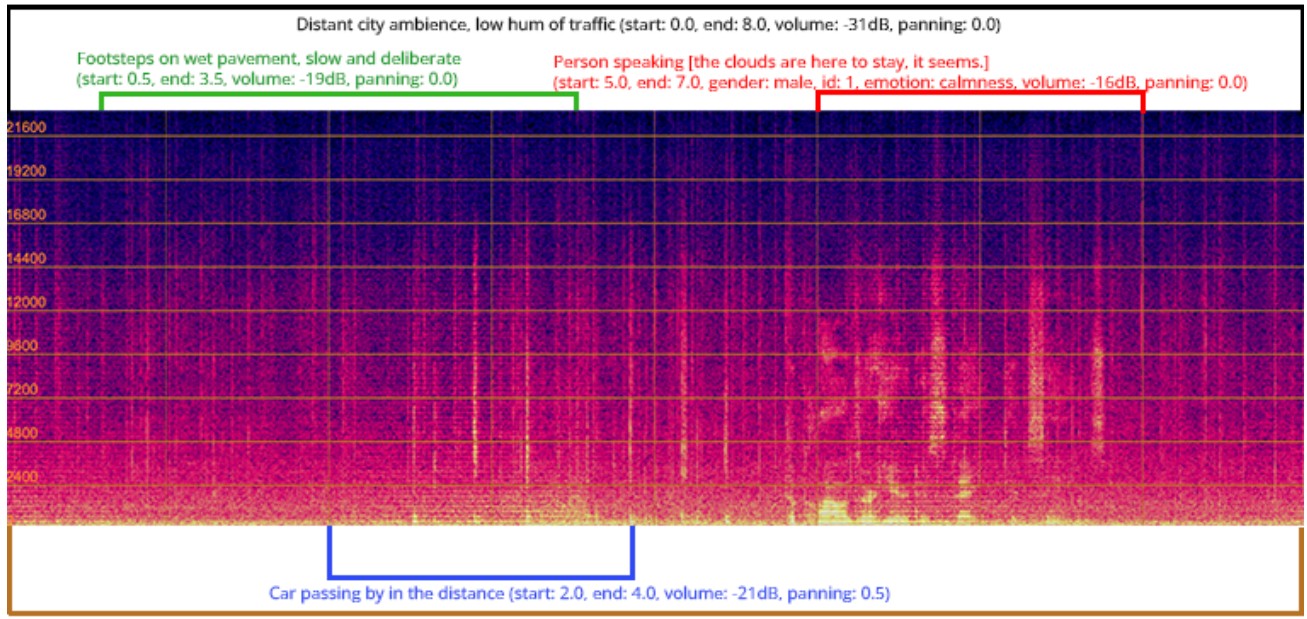

*Figure 7.* Visualization of AudioChat's fine-grained audio captioning capabilities.

- 5: The audio is changed exactly as specified by the text instruction.

We tested 442 audio samples for each system across 139 annotators and obtained a total of 636 ratings per method. The annotators were pre-screened with the following criteria: English speaking, literate, no hearing issues, and good performance in previous unrelated annotation tests.

### E.5. Baseline Methods

We provide an outline and description of our baseline methods in this section.

- StableAudio Open (Evans et al., 2025) and TangoFlux (Hung et al., 2024) are DiT-style T2A models (Peebles & Xie, 2023). The generated latents are obtained from a VAE (Kingma & Welling, 2014) and conditioned on text encoder outputs (like T5). They serve to ground our models' results with that of strong public SOTA models.

- We also train our own DiT model on the same data as AudioChat for fairness and to evaluate audio editing performance of DiTs.

We train 3 LLM-based models: Dense, MoT, and SCT (proposed). All are initialized from Gemma 2 2B (Gemma Team et al., 2024). All models we train use the same training data and VAE.

- The Dense architecture is identical to Gemma, except with the inclusion of the diffusion head and audio input projection.

- The MoT (Liang et al., 2025) model is a decoder-only LLM with 5B parameters. Gemma's Transformer backbone is duplicated to initialize the generation expert's weights. The original weights serve as the understanding expert.

- The SCT (proposed) has 3.6B parameters. The U understanding layers correspond exactly to Gemma. The G generation layers are initialized from scratch. We use G=U / 2, which we found to perform similarly to a heavier G=U setup from early training curves.

```
You are an intelligent AI prompt engineer.
Your job is to simulate interactions between a user and an AI Sound Designer that helps
create sound effects. The AI sound designer is helping the user generate the information
needed for audio editing.

I will give you a single sentence to inspire your simulations, you do not have to follow
it closely. Please be creative in your interpretation of the inspiration.
Your responses should consist of a JSON code block containing up to 5 interactions between
the user and the AI sound designer, for a total of 10 messages. Only return a single JSON.
Each message from the AI-designer should contain up to 10 prompts (one should be ambient),
and other parameters for the audio generation.

Before generating the prompt in each turn, include a natural language description of the
entire soundscape. Don't include any references to loudness or panning in the description.

The AI sound designer has access to one sound effects tool:
- sfx: used to create non-speech sound effects, including music or ambience. You will
provide the text caption of the sound to be genereated.

You will also provide the following information for each prompt:

tool:
- which tool should be used (sfx)
text:
- the audio caption for SFX
loudness:
- How loud the sound should be in decibels
- Use any float between -30 and -10
- Higher loudness should be used for prominent foreground elements (-12).
- Lower loudness should be used for ambient sounds (e.g -30 dB).
panning:
- How much to pan the sound left or right
- Use any float between -1.0 (hard left) and 1.0 (hard right)
- Ambience should be centered (-0.1 to 0.1)
start_time:
- When the sound should start playing in seconds
- Use 0.0 for ambient sounds and long background elements
- Use different start times (0.0-3.0) to sequence shorter sound effects
- Keep start_time between 0 and 4 seconds
- the maximum start time for any sound is 4 seconds.
duration
- how long the sound should play in seconds
- start_time + duration should be <= 8 seconds for all sounds. THIS IS VERY IMPORTANT.
- ambient sounds should always be present (start_time + duration for all non-ambient
↪   sounds should be <= ambience duration)

A unique integer id should be assigned to each prompt.
Whenever a sound is removed, add it to a list of removed ids. Whenever a sound is added,
↪   add it to a list of added ids. Whenever the volume, panning, start time, or duration
↪   of a sound is changed, add it to a list of changed ids.

Example inspiration: leaves rustle
Example output: <see Figure 8 and 9>
```

*Figure 8.* System prompt for AudioCopilot when generating scenes without speech.

```json
[
    {
        "role":"user",
        "content":[
            {
                "type":"text",
                "text":"Create a soundscape of a quiet, refined morning in a
                luxurious room."
            }
        ]
    },
    {
        "role":"system",
        "content":[
            {
                "type":"json",
                "description":"A peaceful, elegant morning soundscape within a
                luxurious room. Delicate sounds of birdsong and a gentle ticking
                clock create a serene atmosphere. The faint aroma of fresh flowers
                lingers in the air.",
                "added":[],
                "changed":[],
                "removed":[],
                "text":[
                    {
                        "tool":"sfx",
                        "text":"gentle birdsong outside a window",
                        "loudness":-28,
                        "panning":0.1,
                        "start_time":0,
                        "duration":8,
                        "id":0
                    },
                    {
                        "tool":"sfx",
                        "text":"antique clock ticking softly",
                        "loudness":-30,
                        "panning":-0.1,
                        "start_time":0,
                        "duration":8,
                        "id":1
                    },
                    {
                        "tool":"sfx",
                        "text":"distant, refined chatter",
                        "loudness":-30,
                        "panning":0,
                        "start_time":0,
                        "duration":8,
                        "id":2
                    }
                ]
            }
        ]
    }
]
```

21

*Figure 9.* An example of the audio story JSON generated by AudioCopilot

```
[
    {
        "role":"user",
        "content":[
            {
                "type":"text",
                "text":"I want to emphasize the delicate nature of the flower
                detail mentioned. Add a sound for someone delicately handling a flower."
            }
        ]
    },
    {
        "role":"system",
        "content":[
            {
                "type":"json",
                "description":"A peaceful, elegant morning soundscape within a luxurious
                room. Delicate sounds of birdsong, a gentle ticking clock,
                refined chatter, and someone delicately handling a flower create
                a serene atmosphere. The faint aroma of fresh flowers lingers
                in the air.",
                "added":[3],
                "changed":[],
                "removed":[],
                "text":[
                    {
                        "tool":"sfx",
                        "text":"delicate flower petals rustling",
                        "loudness":-22,
                        "panning":0,
                        "start_time":1.2,
                        "duration":2.5,
                        "id":3
                    }
                ]
            }
        ]
    }
]
```

*Figure 10.* An example of the audio editing JSON generated by AudioCopilot. We omit the previously generated sound effect and only show the added sound effect due to lack of space. In practice, AudioCopilot will always show all sound params that are present in the scene.

