# OpenReview forum: "AudioChat: Unified Audio Storytelling, Editing, and Understanding with Transfusion Forcing"
_ICML.cc/2026/Conference — ICML 2026 regular_

### Official Review · Reviewer_jvT2 · 2026-02-28

**Soundness:** 3
**Presentation:** 3
**Significance:** 3
**Originality:** 3
**Overall Recommendation:** 4
**Confidence:** 2

**Summary:**

This paper introduces AudioChat, a unified framework designed to handle a range of audio tasks, including editing, understanding, and generation. To overcome the challenge of data scarcity, the authors propose a novel pipeline named AudioCopilot for synthetic data generation. The framework further integrates a series of targeted modifications to multimodal LLMs, such as diffusion forcing to prevent training degradation and customized attention masks for handling mixed-modality inputs. Experimental results demonstrate that AudioChat achieves superior performance across multiple tasks, outperforming strong baseline models.

**Compliance With Llm Reviewing Policy:**

Affirmed.

**Key Questions For Authors:**

The concept of SCT is not very clear. Does it mean the whole proposed network framework? If so, the ablation study is insufficient, as it lacks an evaluation of the individual contributions from key modules like diffusion forcing and the modified attention mask.

**Limitations:**

Please refer to the weakness and questions.

**Strengths And Weaknesses:**

Strength:
1. The proposed AudioChat incorporates a series of well-designed modules that successfully adapt multimodal LLMs to audio-related tasks. It offers meaningful insights for related research areas, for example, the diffusion forcing mechanism helps prevent the model from simply copying the input, encouraging more robust generation.
2. AudioCopilot effectively addresses the long-standing challenge of data scarcity in audio domains by generating substantial amounts of synthetic data.
3. The experimental evaluation is comprehensive and promising, encompassing both automatic metrics and human evaluation, which strengthens the validity of the findings.
Overall, this paper presents an interesting and solid contribution, offering sufficient insights to advance related fields.

Weakness:
1. While AudioCopilot generates large volumes of synthetic data, the discussion on data quality is notably lacking. Only a brief description is provided in Appendix A3.
2. The proposed diffusion forcing mechanism appears to be particularly beneficial for overcoming challenges in multi-turn dialogue. A more detailed experimental analysis would be benefitial.
3. I suggest that the authors provide a more detailed description of the ablation methods in the appendix, clearly highlighting the differences between these baselines and the proposed architecture.
4. It appears that certain necessary ablation studies are missing. For instance, an analysis of the proposed attention masking mechanism would help clarify its contribution to the overall performance.

---

> ### Author Rebuttal · Authors · 2026-03-31
>
> Thank you for your review. We responded to each point below.
>
> ### Weaknesses
>
> 1. Apologies for the lack of clarity. We will add more information as follows:
> - We found that modern LLMs were sufficiently strong at generating realistic scenes in a one-shot manner without much prompt tuning. However, their performance was inconsistent at the O(10 million) scale. As such, we implemented several prompts to ensure consistent generation quality.
>     - Seeding from pre-existing text data (ASR transcripts, audio captions, etc.) was crucial for generating diverse scenes. Changing only the RNG seed with a fixed prompt led to small variations in the same few scenarios. This behavior has been studied in concurrent work on the lack of randomness in LLMs [2].
>     - Outlining exact boundaries for mixing parameter values, like -1.0 is hard left panning or -30dB is appropriate loudness for ambience. Otherwise, the LLM may generate objects that are too loud or too quiet.
>     - Enforcing that ambience must be present throughout the entire audio. Otherwise, certain sounds may "hang" in isolation near the end of the audio.
> - We do not explicitly control for audio quality for cases where the T2A or TTS model may have failed (such as WER or CLAP filtering). These were very rare (upon a manual analysis of 100+ generated samples) and we believed could actually be helpful in certain editing cases (a sound could be changed even if the model mistakenly mis-captions one sound as another during inference). The samples used as editing input in the demo page are well-representative of the overall generated data.
> - The main quality control was through JSON-level validation. We accomplished this by forcing the model to output which element IDs were added, removed, or changed across each editing turn (see Figures 9 and 10). We then compare those element IDs with the content of the generated JSON, filtering out any mismatches. This process removed over 80% of the generated JSONs. Early models trained without the filtering could perform editing well (editFLAM of ~17) but yielded poor consistency scores (20+ multiFLAM). As such, we adopted the pre-train + fine-tune approach discussed in 4.3 to leverage both large-scale weak data and the small high quality subset.
>
> 2. Reviewer k1M4 raised a similar point. Please refer to our response above under section _Multi-turn Editing_
>
> 3. We will highlight here the main differences between the ablation methods. This will be included with more detail in the appendix in future drafts.
>  - StableAudio Open (SAO) and TangoFlux are DiT-style T2A models. The generated latents are obtained from a VAE and conditioned on text encoder outputs (like T5). They serve to ground our models’ results with that of strong public SOTA models.
> - We also train our own DiT model on the same data as AudioChat for fairness and to evaluate audio editing performance of DiTs.
> - We train 3 LLM-based models: Dense, MoT, and SCT (proposed). All are initialized from Gemma 2 2B.
> - The Dense architecture is identical to Gemma, except with the inclusion of the diffusion head and audio input projection.
> - The MoT model is a decoder-only LLM with 5B parameters. Gemma’s Transformer backbone is duplicated to initialize the generation expert’s weights. The original weights serve as the understanding expert.
> - The SCT has 3.6B parameters. The U understanding layers correspond exactly to Gemma. The G generation layers are initialized from scratch. We use G=U / 2, which we found to perform similarly to a heavier G=U setup from early training curves.
> - All models we train use the same training data and VAE.
>
> 4. We will make sure to include this ablation for the camera ready. For reference, we observed that models trained only using causal attention generated semantic content well (intelligible speech or realistic sound effects) but with poor audio fidelity (noise, distortions, artifacts). Changing to the proposed method significantly improved this. In the meantime, we refer reviewers to Section 4.3.1 of [1], which showed significant improvements with the same attention approach in image generation fidelity (FID) and little change in semantics (CLIP score).
>
> ### Questions
>
> > The concept of SCT is not very clear. Does it mean the whole proposed network framework?...
>
> Apologies for the confusion, SCT only refers to our model's neural architecture (using the first $U$ layers for understanding and the next $G$ layers for generation). All LLM-based models in the ablations (Dense, MoT, SCT) are trained with both diffusion forcing and the modified attention mask. As such, the ablated architectures are fully comparable.
>
> [1] Zhou, Chunting, et al. "Transfusion: Predict the next token and diffuse images with one multi-modal model." ICLR 2025.
>
> [2] Zhao, Minda, Yilun Du, and Mengyu Wang. "Large Language Models Are Bad Dice Players: LLMs Struggle to Generate Random Numbers from Statistical Distributions." arXiv preprint arXiv:2601.05414 (2026).

---

> > ### Author Rebuttal · Reviewer_jvT2 · 2026-04-02
> >
> > Thank the authors for the response. All my concept concerns are get fully addressed. However, the added ablation study lacks of quantitative metrics. Therefore, I maintain my score.

---

### Official Review · Reviewer_T3NH · 2026-03-09

**Soundness:** 4
**Presentation:** 3
**Significance:** 4
**Originality:** 3
**Overall Recommendation:** 5
**Confidence:** 3

**Summary:**

This paper intends to unify audio editing, storytelling, and understanding into a single pipeline. They proposed AudioCopilot, which uses a pre-trained LLM (GPT -3 27B) to simulate multiturn interactions between a user and an AI sound designer. After collecting synthesized data, they built AudioChat with an audio tokenizer and a multi-modal language model to achieve the proposed three tasks. For evaluation, they proposed multiFLAM, editFLAM, and ∆multiFLAM to measure the ability of storytelling, editing, and understanding.

**Compliance With Llm Reviewing Policy:**

Affirmed.

**Final Justification:**

My concerns have largely been addressed, I will raise the score

**Key Questions For Authors:**

1. Do you think that this model can generalize to real-world data?
2. Will the model or the data generated by AudioCopilot be open-sourced?
3. For the audio editing task, if the original audio is a piece of music, will the added sound be harmonically compatible with the original source?

**Limitations:**

yes

**Strengths And Weaknesses:**

# Strengths
1. Propose a novel framework for a unified multimodal model capable of reasoning and editing audio end-to-end.
2. The situation of containing multiple sources and speakers hasn't been addressed by previous methods
3. Audio from multiple turns is highly related and will cause the model to copy the input latent. The author uses the tranfusion forcing to solve this, which is really interesting.
4. New metrics have been proposed, and they could be used as standard metrics for the three tasks.
5. The architecture ablation is persuasive.

# Weaknesses
1. The training data and evaluation data are all from AudioCopilot, which are all synthetic data. The generalization to real-world audio scenes is a concern.
2. Comparison with other audio foundation models (e.g., AudioFlamingo, Qwen-audio, or UALM ) for audio understanding, although I understand that they might not be able to understand time annotations.
3. Only Audio Editing contains human evaluation.

---

> ### Author Rebuttal · Authors · 2026-03-31
>
> We thank you for the review. We address each point in our response below.
>
> ### Weaknesses
> 1. We added additional results using real audio from AudioCaps as a test set, which showed that the model generalized well. Since Reviewer k1M4 raised the same point, please refer to our response above for more detail.
> 2. We have included comparisons with AudioFlamingo and Qwen2-Audio for audio understanding (UALM is not publicly available). We prompt the models to 1.) output a list of all elements occurring in the audio without timestamps and 2.) perform diarization with transcriptions and timestamps. We measure multiFLAM on the former and tcpWER on the latter.
>
> | Model | tcpWER | multiFLAM |
> |---|---|---|
> | AudioChat | 9.7 | 86.3 |
> | Qwen 2 Audio | 74.5 | 63.6 |
> | AudioFlamingo 3 | 100+ | 76.6 |
>
> The results show that AudioChat performs better than both models on both tasks. While they performed decent on the multi-source audio captioning task, they could not perform diarization at all. In addition to typical recognition errors, we observed that this was often due to the models' inability to consistently follow instructions. For example, AudioFlamingo could not adhere to a structured output format (such as JSON) despite our attempts at optimizing the input prompt. Qwen Audio would often caption non-speech audio despite being instructed otherwise. Both models frequently added timestamps when not desired or did not output timestamps when desired.
>
> In general, this shows the effectiveness of AudioChat's structured reasoning approach - the model's outputs are easy to evaluate and manipulate for downstream use, allowing it to perform both tasks reasonably well.
>
> ### Questions
> 1. Addressed above.
> 2. The model in its current state will not be open-sourced since it is trained on proprietary data. However, we are currently working on and hope to open source 1.)  the data generated from AudioCopilot and 2.) an AudioChat checkpoint trained using only publicly available data along with re-computed evaluation metrics.
> 3. Generally, the model cannot perform such a fine-grained level of music editing, since we do not train on any music-specific data or supervision. It will treat music as any other audio sound source, so while it can change the style or instrument, there is no guarantee that the melody will remain the same.

---

> > ### Author Rebuttal · Reviewer_T3NH · 2026-04-03
> >
> > Thank you for your response. My concerns have largely been addressed.

---

### Official Review · Reviewer_WZSM · 2026-03-10

**Soundness:** 4
**Presentation:** 2
**Significance:** 4
**Originality:** 4
**Overall Recommendation:** 5
**Confidence:** 4

**Summary:**

This paper proposes AudioChat, a unified audio foundation model which can understand, generate, and edit complex multi-source “audio stories” that include both speech and non-speech sounds at 48 kHz. The core contributions are: AudioCopilot, a tool-calling LLM agent to synthesize audio-story data; Audio Transfusion Forcing, combining Transfusion with Diffusion Forcing scheme and custom attention masking to support interactive editing; and three task-oriented evaluation metrics based on OpenFLAM frame-wise probabilities.

**Compliance With Llm Reviewing Policy:**

Affirmed.

**Final Justification:**

The authors have addressed my concerns and I will keep my initial score.

**Key Questions For Authors:**

How to evaluate the effectiveness and correctness of your generated dataset? Have you ever cleaned the data by human? How does AudioChat perform on real, human-recorded audio editing tasks (e.g., removing a speaker/background or altering spatialization) without access to AudioCopilot-style structured traces? Can you report human MOS and FLAM-based metrics on a small real-world set?

**Limitations:**

I thinke the limitations should be further discussed according to my key questions.

**Strengths And Weaknesses:**

Strengths: An important while difficult problem was proposed and an acceptable solution was given; A complete pipeline including Copilot for data creation, AudioChat for audio-story generation, understanding and editing; Three novel evaluation metrics proposed for this tasks; Sufficient objective and subjective experimenta for validation.

Weakness: The figures do not appear aesthetically pleasing; Limited evaluation on real-world, non-synthetic data for editing and storytelling;.

---

> ### Author Rebuttal · Authors · 2026-03-31
>
> Thank you for your review. We have group our responses by subject area and responded to each below.
>
> > The figures do not appear aesthetically pleasing;
>
> We apologize for the poor quality of the figures. We have increased their resolution and readability. They can be found in this GitHub repo: https://github.com/audiochat-icml-2026/audiochat-figures . If you have any specific comments about the aesthetics, we would be happy to address them.
>
> >  Limited evaluation on real-world, non-synthetic data for editing and storytelling... How does AudioChat perform on real, human-recorded audio editing tasks (e.g., removing a speaker/background or altering spatialization) without access to AudioCopilot-style structured traces? Can you report human MOS and FLAM-based metrics on a small real-world set?
>
> Reviewer k1M4 raised a similar point and we conducted these analyses for editing by creating a new test set from AudioCaps. Please refer to our response above in the _Real World Generalization_ Section for more information.
> - Since audio storytelling is a purely text-conditional audio generation task, it does not need audio inputs and therefore evaluation on real-world audio cannot be performed.
> - Since AudioChat has understanding capabilities, it can generate structured traces from real-world data. These are used for real-world audio editing.
> - We reported above performance with editFLAM and also added new samples for real world editing in our demo page. While we are unable to perform human evaluation on this set within the limited rebuttal period, we may perform it for the camera ready if resources permits.
>
> > How to evaluate the effectiveness and correctness of your generated dataset? Have you ever cleaned the data by human?
>
> Reviewer jvT2 raised a similar question. Please refer to our responses below under Weaknesses Point 1.

---

> > ### Author Rebuttal · Reviewer_WZSM · 2026-04-01
> >
> > Thanks for your rebuttal, I hope that you could add these further clarifications into your final version. Besides, a minor suggestion is that the box lines and arrows in Figure 3 of the final version of the paper could be aligned more neatly.

---

> > > ### Author Response · Authors · 2026-04-02
> > >
> > > Thank you. We will certainly add all of these clarifications and details to the final version. We have also updated figure 3 so that the boxes and arrows are better aligned.
> > >
> > > https://github.com/audiochat-icml-2026/audiochat-figures/blob/main/fig3.pdf

---

### Official Review · Reviewer_k1M4 · 2026-03-12

**Soundness:** 2
**Presentation:** 3
**Significance:** 3
**Originality:** 3
**Overall Recommendation:** 4
**Confidence:** 4

**Summary:**

The paper presents AudioChat, a unified audio model designed to generate, edit, and understand complex multi-source acoustic scenes, referred to as "audio stories." The authors address the limitations of current audio models in handling intricate scenes involving multiple speakers and overlapping foreground/background sound effects. To overcome data scarcity, they introduce AudioCopilot, a tool-calling LLM agent that synthesizes millions of multi-turn conversations for training. Furthermore, they propose a novel training objective, Audio Transfusion Forcing, and a Self-Cascaded Transformer (SCT) architecture to enable structured reasoning.

**Compliance With Llm Reviewing Policy:**

Affirmed.

**Final Justification:**

The authors address some of my concerns. I will increase the score to 4.

**Key Questions For Authors:**

- Can the authors provide or point to specific examples in the demo that showcase a full sequence of multi-turn edits on a single audio scene? How does the model prevent "semantic drift" as the number of editing turns increases?

- In the Self-Cascaded Transformer (SCT), how was the ratio of "Understanding Layers" (U) to "Generation Layers" (K-U) determined? Did the authors experiment with dynamic allocation or different split points?

- How does AudioChat perform when the input audio contains significant environmental noise or low-quality recordings that were not part of the synthetic AudioCopilot distribution?

**Limitations:**

yes

**Strengths And Weaknesses:**

## Strengths

- Successfully unifies diverse tasks (Understanding, Storytelling, and Editing) into one end-to-end model, significantly reducing the latency compared to traditional cascaded agent-based systems.

- The AudioCopilot pipeline is a highly effective solution for the lack of granularly annotated audio-story data, leveraging synthetic dialogues to scale training data to 6 million samples.

- Unlike many existing models limited to 16kHz monophonic audio, AudioChat supports 48kHz stereo and explicitly models spatial characteristics like panning and loudness.

- The use of structured Chain-of-Thought (CoT) allows for interpretable processing, letting users see how the model decomposes abstract prompts into individual sound components.

## Weaknesses

- The paper emphasizes "interactive multi-turn audio understanding/generation" as a core contribution. However, the current online demo lacks comprehensive multi-turn editing examples. Most demonstrations appear to be single-turn or isolated tasks. Seeing the model's ability to maintain narrative consistency across 3+ turns of editing is crucial for verifying the "Transfusion Forcing" claims.

- Several figures throughout the manuscript (e.g., Figure 1 and Figure 3) suffer from low resolution and blurry text. The "Structured Reasoning Trace" in Figure 1 and the architectural labels in Figure 3 are difficult to read, which hinders the clarity of the technical presentation.

- The model is primarily trained on synthetic data generated by AudioCopilot. While the scale is impressive, synthetic data often lacks the nuanced "imperfections" of real-world multi-source recordings (e.g., natural reverberation, complex overlapping speech). There is limited evidence showing how well the model generalizes to non-synthetic, raw audio inputs.

- The proposed evaluation metrics (multiFLAM, editFLAM) rely heavily on the OpenFLAM model. If the underlying embedding model has biases or lacks sensitivity to specific sound classes, the resulting evaluation of AudioChat may be skewed.

The current illustrations reduce the paper's professionalism; I would raise the grade if the author could improve them.

---

> ### Author Rebuttal · Authors · 2026-03-31
>
> Thank you for your comments. We address each point below. All of these analyses will be added to future drafts.
>
> ### Multi-turn Editing
>
> Thank you for pointing this out. We show below results on 3-turn dialogue. The model is input with 2 audio-text pairs along with the final editing instruction.
>
> It can be visualized as `[instruction 1] [audio 1] [instruction 2] [audio 2] [instruction 3]`.
>
> We show results for two cases:
> 1. Using the ground truth audio from AudioCopilot (topline)
> 2. Use the model's actual output for audio 2 (when given  `[instruction 1] [audio 1] [instruction 2]`.)
>
> | Audio Context | $\triangle$multiFLAM ($\downarrow$)| EditFLAM ($\uparrow$)|
> |---|---|---|
> | Ground Truth (topline) | 10.5 | 11.1 |
> | Inference Result | 11.8 | 12.4 |
>
> Our evaluation shows that degradation from using actual edit outputs is fairly minimal (1.3% absolute in both metrics). We attribute this to the model's long-form multi-turn context - access to the original input helps mitigates mistakes from intermediate outputs and therefore semantic drift.
>
> We have included example multi-turn outputs in the demo page.
>
> ### Figures
>
> We apologize for the poor quality of the figures. We have increased their resolution and readability. They can be found in this GitHub repo: https://github.com/audiochat-icml-2026/audiochat-figures .
>
> ### Real World Generalization
>
> We agree that this is an important evaluation. We test this by creating a test set from the AudioCaps test set, which contains noisy audio sourced from YouTube. We use prompt an LLM (Qwen3 35B) to generate an edit instruction given the caption and one of the edit operations (add, remove, change, etc). We then calculate editFLAM on model outputs from these audio+instruction pairs (we cannot calculate $\triangle$multiFLAM since it requires before-after captions).
>
> | Audio Source | EditFLAM |
> |---|---|
> | Synthetic | 18.6 |
> | Real | 15.5 |
>
> While there is indeed a drop in performance, we believe the drop is fairly small (3%) for a model trained solely on synthetic audio, indicating that our approach generalizes well. We have included examples in the demo page.
>
> In general, we observed that errors generally occur when there are severe mistakes in the generated reasoning trace that differ too much from the input audio ($P(X^{cap}\_{t-1}|Y_{t-1})$ in Eq. 2).
> - The reasoning trace includes something not in original and the model generates it
> - The reasoning trace misses something in original, and the model cannot locate it and therefore does not edit correctly.
>
> Both cases can likely be addressed in future work by improving the audio understanding capabilities of the model, such as with real-world captioning data or more training data (we only use ~13.3k synthetic hours).
>
> ### Evaluation Metrics
>
> We agree that evaluation may be skewed if OpenFLAM may have biases towards/against certain sound classes. However, this a weakness for any embedding-based metric, including the commonly used CLAPscore or FAD. One method to mitigate this is to use multiple distinct embedding models. We re-calculate here our editing metrics, but with the Perception-Encoder Audio (PE-A) used by SAM-Audio instead of OpenFLAM. The best method is marked in **bold** and the second best in _italic_.
>
> | | $\triangle$multiFLAM (↓) | editFLAM (↑) |$\triangle$PE-A (↓) | edit PE-A (↑) |
> |---|---|---|---|---|
> | Ground Truth | **8.2** | **19.4** | **2.7** | **23.6** |
> | DiT | 12.5 | 2.7 | 4.1 | 4.0 |
> | Diffusion LLM | 27.8 | 18.1 | 13.2 | 18.7 |
> | Cascade | 26.7 | 17.6 | 19.1 | 23.1 |
> | AudioChat | _11.7_ | _18.6_ | _3.0_ | _23.3_ |
>
> The rankings and relative score using PE-A are consistent with both FLAM-based metrics and human evaluation - the ground truth is correctly ranked as the top line, AudioChat is clearly the best model, DiffusionLLM and Cascade have similar performances, and the DiT is clearly only copying the input. This shows that our rankings and proposed metrics are robust and still correlate well with human judgement when using a different embedding model.
>
> ### Questions
>
> > Semantic drift in multi-turn edits
>
> See above.
>
> > SCT Settings
>
> The U understanding layers correspond exactly to Gemma. We use G=U / 2, which we found to perform similarly to a heavier G=U setup while being more efficient from early training curves. We did not experiment with dynamic allocation, although we believe it would be promising future work.
>
> > Performance with environmental noise or low quality recordings
>
> Environmental noise or low recording quality in real world audios did not significantly affect performance. We attribute the former to many of our simulations containing significant non-speech audio or ambience. For the latter, there are also many "low-quality-style" simulations due to inconsistencies in the T2A/TTS models used. Most editing errors came from incorrect captions, which generally occurred due to difficult acoustic conditions (overlapping / unintelligible speech) or ambiguous sound sources.

---

> > ### Author Rebuttal · Reviewer_k1M4 · 2026-04-06
> >
> > Thank you for your response. I will increase my score to 4.

---

### Decision · Program_Chairs · 2026-04-30

**Decision:**

Accept (regular)

**Comment:**

This paper proposes a unified multimodal (text+audio) framework for handling diverse audio tasks, including (multi-turn) audio editing, storytelling generation, and understanding, based on an LLM along with a diffusion-based audio decoder. The method leverages a tool-calling LLM agent to generate large-scale synthetic training data, and adopts a transfusion forcing objective with custom attention masks and multi-stage training to enable multimodal learning.

Overall, the proposed framework seems to be technically sound and presents a first promising attempt to unify diverse audio tasks, including understanding, generation, and multi-turn editing, within a single framework. Moreover, by leveraging the internal reasoning capability of an LLM, the approach offers a meaningful contribution. The authors also empirically demonstrate that, despite being a unified framework, the proposed method achieves strong performances.

Many concerns raised by the reviewers are addressed in the rebuttal, including the lack of real-audio experiments, potential bias in evaluation metrics, validation of the synthetic training data, and comparisons with other audio foundation models for audio understanding.

Therefore, based on the overall positive assessments from the reviewers, I recommend acceptance of this paper.

However, there are still unresolved concerns that need be addressed in the revised version.

- The authors claim that the performance drop on real audio editing (AudioCaps) is relatively small (3%). While the absolute decrease is 3.1, this corresponds to a ~16.6% relative drop, which is non-negligible. This gap could become more pronounced in more complex settings, such as multi-turn scenarios. More extensive experiments and a deeper validation of the proposed framework are needed on real-world generalization.

- From an architectural perspective, the paper provides comparisons with existing models and includes some ablation studies. However, ablations on other important components remain limited. For instance, it would be valuable to evaluate the transfusion objective without diffusion forcing. In addition, further analysis is needed to assess the impact of the customized attention masks, as well as the contribution of each training stage through more comprehensive ablation studies.